# Trajectory-guided Control Prediction for End-to-end Autonomous Driving: A Simple yet Strong Baseline

**Penghao Wu**[*†]
Shanghai AI Laboratory
Shanghai Jiao Tong University
wupenghaocraig@sjtu.edu.cn

**Xiaosong Jia**[*]
Shanghai Jiao Tong University
Shanghai AI Laboratory
jiaxiaosong@sjtu.edu.cn

**Li Chen**[*]
Shanghai AI Laboratory
lichen@pjlab.org.cn

**Junchi Yan**[†]
Shanghai Jiao Tong University
Shanghai AI Laboratory
yanjunchi@sjtu.edu.cn

**Hongyang Li**
Shanghai AI Laboratory
Shanghai Jiao Tong University
lihongyang@pjlab.org.cn

**Yu Qiao**
Shanghai AI Laboratory
qiaoyu@pjlab.org.cn

## Abstract

Current end-to-end autonomous driving methods either run a controller based on a planned trajectory or perform control prediction directly, which have spanned two separately studied lines of research. Seeing their potential mutual benefits to each other, this paper takes the initiative to explore the combination of these two well-developed worlds. Specifically, our integrated approach has two branches for trajectory planning and direct control, respectively. The trajectory branch predicts the future trajectory, while the control branch involves a novel multi-step prediction scheme such that the relationship between current actions and future states can be reasoned. The two branches are connected so that the control branch receives corresponding guidance from the trajectory branch at each time step. The outputs from two branches are then fused to achieve complementary advantages. Our results are evaluated in the closed-loop urban driving setting with challenging scenarios using the CARLA simulator. Even with a monocular camera input, the proposed approach ranks *first* on the official CARLA Leaderboard, outperforming other complex candidates with multiple sensors or fusion mechanisms by a large margin. The source code is publicly available at https://github.com/OpenPerceptionX/TCP.

## 1 Introduction

End-to-end autonomous driving methods, which directly map raw sensor data to a planned trajectory or low-level control actions, show the virtue of simplicity, conceptually avoiding the cascading error of complex modular design and heavy hand-crafted rules. The output prediction of the model for end-to-end autonomous driving generally falls into two forms: trajectory/waypoints [48, 4, 11, 46, 15, 29, 10] and direct control actions [17, 39, 18, 42, 12, 60, 9]. However, there is still no clear conclusion as to which of these two forms is better for all circumstances or certain scenarios.

Different from control predictions that could be directly applied to the vehicle, for methods that **plan trajectory**, additional controllers such as PID controllers are usually needed as a subsequent step to convert the planned trajectory into control signals. One attractive and potential supremacy of trajectory-based prediction is that it actually considers a relatively longer time horizon into the future and could be further combined with other modules (*e.g.*, multi-agent trajectory prediction [59, 10],

---

[*] Equal Contribution. Work done when PW and XJ were interns at Shanghai AI Laboratory.

[†] Correspondence author.

36th Conference on Neural Information Processing Systems (NeurIPS 2022).

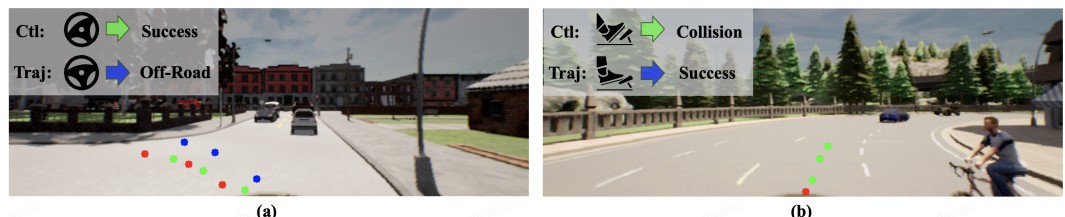

Figure 1: Typical failure cases of two prediction paradigms. Red dots indicate the trajectory prediction, blue dots are the actual path following the red trajectory with PID controllers, and green dots denote the actual path from control-based method. (a) trajectory-based methods may struggle for big turns. (b) control-based methods may have a reaction latency and suffer from abrupt obstacles due to focusing on current time step only. These observations motivate us to propose a unified framework to combine these two worlds for mutual benefits.

semantic or occupancy prediction modules [15, 8, 25]) to reduce possible collisions. However, turning the trajectory into control actions so that the vehicle could follow the planned trajectory is not trivial [57]. The industry usually adopts sophisticated control algorithms such as model predictive control to achieve reliable trajectory-following performance [5, 21]. Simple PID controllers may perform worse in situations such as taking a big turn or starting at the red light due to the inertial problem of end-to-end models [29]. For **control-based** methods, the control signals are directly optimized. Nevertheless, their focus on the current step may cause deferred reactions to avoid potential collisions with other moving agents. The independence between the control predictions of different steps also makes the actions of the vehicle more unstable or discontinuous. Fig. 1 shows two typical cases where two paradigms fail respectively. How to combine these two forms of prediction model as well as their outputs is an interesting yet relatively rarely studied area, which motivates this work.

One straightforward (but in fact rarely studied in literature) idea is to train a control prediction model and a trajectory planning model separately, and combine their ultimate outputs directly. It can be viewed as an ensemble of two different models. However, such a naive approach not only doubles the size of the model, but also ignores possible useful correlations between these two forms. To this end, we introduce the **TCP** (Trajectory-guided Control Prediction) framework, packing these two branches into a unified framework. It can be viewed as a multi-task learning (MTL) [7, 2] framework where a shared backbone extracts common features with decreased computational complexity as well as the increased ability of generalization due to the close relationship between the two tasks [38, 34, 13]. Furthermore, to address the drawbacks of current control prediction methods, we delicately devise a novel multi-step control branch and a trajectory-guided control prediction scheme.

While trajectory planning considers several steps into the future, directly learning the control in a behavior cloning fashion [44, 41, 17, 18, 11] often focuses on the current time step only, given prior on each state-action pair as independent and identical distributed (IID). This assumption is not accurate and may hamper the long-term performance since the driving task is a sequential decision-making problem. To alleviate the problem, we propose to predict multi-step control actions into the future. However, the multi-step control process needs interactions with the environment. Thus we formulate a temporal module to learn the forward process and interactions between the ego agent and the environment. A temporal module implemented with GRU [16] progressively deals with the feature representation for each time step, implicitly taking into account the dynamic motion of agents, interaction among them and dynamic environment information such as the changing of traffic lights.

Additionally, to generate accurate control signals in the multi-step prediction scheme, the model should retrieve proper location information from current sensor input for different future time steps. For example, an agent may pay more attention to nearby regions for a few early future time steps and far away regions for the remote ones. Considering that the knowledge has already been partly encoded in the trajectory branch, we adopt the attention mechanism to locate those critical and helpful areas in the long-term trajectory prediction branch, and guide the control prediction branch to pay attention to them at each future step in a corresponding way. As a result, our model is capable of reasoning about how to optimize current control prediction so that the future states are similar to those from the expert when the predicted control actions are applied.

With the predicted trajectory and control signals from two branches, we propose a situation based fusion scheme to adaptively combine these two forms in a self-ensemble way to form the ultimate output according to the experiments results and prior knowledge. It combines the best of these two forms, which further boosts the performance under different scenarios.

TCP has shown superior performance when being validated in the CARLA driving simulator [20]. Our method, which only uses a **monocular camera**, achieves a **75.137** driving score and ranks **1st** on the public CARLA Leaderboard [1], even surpassing prior state-of-the-art methods using multiple cameras and a LiDAR by 13.291 points. The main **contributions** of this paper include:

- We examine two dominant paradigms for end-to-end autonomous driving: trajectory planning and direct control, and propose to combine them in an integrated learning pipeline. To our knowledge, this is the first time that such two branches are jointly learned and fused for prediction.

- A multi-step control prediction branch with a temporal module and trajectory-guided attention is devised to enable temporal reasoning. To combine the best of two branches, we design a situation based scheme to fuse the two outputs.

- As a simple yet strong baseline, our method with only a monocular camera as input achieves new state-of-the-art on the CARLA Leaderboard with many competitors using multiple sensors. We conduct thorough ablation studies to verify the effectiveness of our approach.

## 2 Related Work

### 2.1 End-to-end Autonomous Driving

Learning-based end-to-end autonomous driving has emerged as an active research topic in recent years. Studies usually fall into two categories: reinforcement learning (RL) and imitation learning. RL is a promising way to address the problem of being more robust to the distribution shifts of datasets. Liang *et al.* [39] use DDPG to train a policy which is pre-trained in a supervised way. Kendall *et al.* [30] train their deep RL algorithm onboard to efficiently learn to drive a real-world vehicle. The perception task is decoupled out of the online RL process in [50, 9, 62]. The model-based method WoR [12] assumes world on rails and uses policy distillation to realize powerful performance.

Imitation learning (IL), especially behavior cloning, collects recorded data for models to mimic with high data efficiency. The expert data typically has two forms, trajectories and control actions. Zeng *et al.* [58] train a cost volume to generate the planning route, while [49, 8, 25] explicitly design safety and comfort costs based on semantic occupancy maps to select the best one in the expert trajectory sets. Zhang *et al.* [59] predict trajectories of surrounding vehicles with labeled BEV map. LBC [11] and NEAT [15] decode waypoints from a dense heatmap or offset map. These approaches aforementioned all utilize a relatively dense representation to obtain results which increases model complexity. Transfuser and its variants [46, 29] adopt a simple GRU to auto-regress waypoints. LAV [10] adopts a temporal GRU module to further refine the trajectory. They unanimously achieve impressive performance on the CARLA leaderboard, motivating us to adapt the auto-regression scheme as well in our design. On the other hand, all trajectory-based methods use PID controllers to get the ultimate actions, which may cause inferior effects in complicated scenarios.

Another genre to predict control actions directly is proposed in [44, 40, 3, 54]. CIL [17] adds a measurement encoder and multiple branches for different high level commands with the image encoder. CILRS [18] is proposed afterwards and further introduces a speed prediction head. They stand as classic baselines for IL in the CARLA driving simulator. Diverse optimized approaches are presented based on them, such as multi-modal inputs [22, 53], multi-task learning [56, 37, 24, 27, 31, 26, 63], dataset aggregation [45] and knowledge distillation [61, 60]. However, the compact control-based methods often have higher vehicles collision rates, remaining an interesting domain to explore. Similar work exists in other related domains such as robotic navigation as well. [43] learns a controller after a local trajectory planner to improve the overall navigation behavior.

### 2.2 Multi-task and Ensemble Learning for Autonomous Driving

Multi-task learning is a popular approach to train several related tasks simultaneously to help each other and improve generalization [7, 2]. Combinations of various autonomous driving tasks such

as object detection, lane detection, semantic segmentation, depth estimation, *etc.* have been proved to be capable of achieving incredible performance [38, 14, 47, 34, 52, 13]. MTL is also suitable in the end-to-end problem since it is observed the performance of a direct mapping from an image to control signals is limited. [56] adds a speed prediction task similar to CIL [17] and [63] separates the lateral and longitudinal controls as two tasks. LAV [10] trains an extra scene mapping network, and [24, 27, 29] additionally predict optical flow or dense depth. Our idea of training trajectory and control simultaneously is closely related to FASNet [31]. FASNet predicts future positions of the ego agent as an auxiliary task and adds a kinematic loss considering the relation between control and locations. However, the constrain is based on a constant velocity model which neglects the important throttle and brake, and it does not work at the inference time. On the other hand, our TCP framework has feature interactions at an earlier stage to fully explore their potential mutual benefits.

Ensembles of models have long been utilized to improve the performance in computer vision [19, 33, 35, 51, 55, 45]. Besides the normal combination of models, two classic ensemble learning methods are particularly preferred in the autonomous driving regime. One is the Test-Time Augmentation (TTA), which is of great help to the 3D object detection task with LiDAR [6, 36]. Another one is the fusion of experts [28] where experts are trained on a subset of the input space and a gating network is trained to provide the fusion weights. LSD [42] and MoDE [32] divide a dataset into sub-scenarios to get different sub-policies for end-to-end autonomous driving. These traditional ensemble approaches combine models of the same structure while our approach tries to combine two different representations. Also, the multiple experts design increases the complexity of the training strategy and we seek to have a simpler situation based fusion scheme to boost the performance.

## 3 Trajectory-guided Control Prediction

### 3.1 Problem Setting

**Problem formulation.** Given the state $\mathbf{x}$ comprised of the sensor signal $\mathbf{i}$, the speed of the vehicle $v$, and the high level navigation information $\mathbf{g}$ including a discrete navigation command and the coordinates of navigation target provided by the global planner, the end-to-end model needs to output control signals $\mathbf{a}$ comprised of longitudinal control signals *throttle* $\in [0, 1]$ and *brake* $\in [0, 1]$, and the lateral control signal *steer* $\in [-1, 1]$.

Conventional methods tackle this problem with either a trajectory-output or a control-output only model. However, TCP combines both of them as two branches: a trajectory branch which predicts the planned trajectory and a control branch which is guided by the trajectory one and outputs both current and multi-step control signals into the future. Both branches are trained in a supervised manner. Consider an expert which directly outputs the control signals at each step, supervising the predicted trajectory with the ground truth trajectory makes it not strictly satisfy the setting of behavior cloning in imitation learning. The ground truth trajectory indeed involves future expert actions and future states about the environment, so we formulate it as a trajectory planning task with ground truth trajectory as supervision for our trajectory branch. As for the control branch, training a control model which makes current control prediction supervised by the expert control is just behavior cloning in imitation learning, and it can be formulated as:

$$\arg \min_{\theta} \mathbb{E}_{(\mathbf{x}, \mathbf{a}^*) \sim \mathrm{D}}[\mathcal{L}(\mathbf{a}^*, \pi_\theta(\mathbf{x}))], \tag{1}$$

where $D = \{(\mathbf{x}, \mathbf{a}^*)\}$ is a dataset comprised of state-action pairs collected from the expert. $\pi_\theta$ denotes the policy of the control branch, and $\mathcal{L}$ is the loss measuring how close the action from the expert and the action from our model is. The expert collects the dataset by controlling the vehicle and interacting with the world. Each collected route is a trajectory $\xi = (\mathbf{x}_0, \mathbf{a}_0^*, \mathbf{x}_1, \mathbf{a}_1^*, \cdots, \mathbf{x}_T)$ as a sequence of state action pairs $\{(\mathbf{x}_i, \mathbf{a}_i^*)\}_{i=0}^T$, which is then added into the whole dataset $D$.

**Expert demonstration.** Here we choose Roach [60] as the expert. Roach is a simple model trained by RL with privileged information, including roads, lanes, routes, vehicles, pedestrians, traffic lights, and stops, all being rendered into a 2D BEV image. Such a learning-based expert can transfer more information besides the direct supervision signals compared with an expert made by hand-crafted rules. Specifically, we have a feature loss which forces the latent features before the final output head from the student model to be similar to that of the expert. A value loss is also added as an auxiliary task for the student model to predict an expected return.

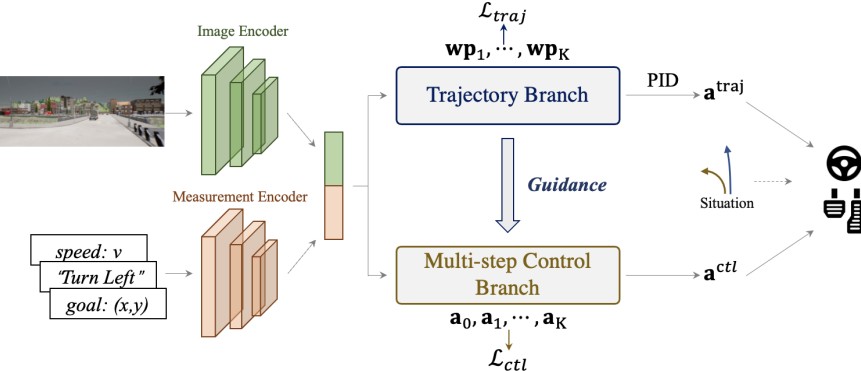

Figure 2: Overview of Trajectory-guided Control Prediction (TCP). The encoded features are shared by the trajectory and multi-step control branch. The trajectory branch provides per-step guidance for multi-step control prediction. Outputs from two branches are combined according to our situation based fusion scheme to generate the ultimate control actions.

## 3.2 Architecture Design

**Overview.** As illustrated in Fig. 2, the whole architecture is comprised of an input encoding stage and two subsequent branches. The input image $\mathbf{i}$ goes through a CNN based image encoder, such as ResNet [23], to generate a feature map $\mathbf{F}$. In the meantime, the navigation information $\mathbf{g}$ is concatenated with the current speed $v$ to form the measurement input $\mathbf{m}$, then an MLP based measurement encoder takes $\mathbf{m}$ as its input and outputs the measurement feature $\mathbf{j}_m$. The encoded features are then shared by two branches for subsequent trajectory and control predictions. Specifically, the control branch is a novel multi-step prediction design with guidance from the trajectory one, which will be illustrated in detail in the following sections. Finally, a situation based fusion scheme is adopted to combine the best of the two output paradigms. We will go over each part in detail below.

### 3.2.1 Trajectory planning branch

Different from control prediction which directly predicts control actions, the trajectory planning branch first generates a planned trajectory comprised of waypoints at $K$ steps for the agent to follow, and then the trajectory is processed by low-level controllers to get the final control actions. With the shared feature from the input encoder, the image feature map $\mathbf{F}$ is average pooled and concatenated with the measurement feature $\mathbf{j}_m$ to form $\mathbf{j}^{\mathrm{traj}}$. Inspired by [46], we feed $\mathbf{j}^{\mathrm{traj}}$ into a GRU [16] to auto-regressively obtain future waypoints one by one to form the planned trajectory altogether.

We have two PID controllers for longitudinal and lateral control respectively. With the planned trajectory, we first calculate the vectors between consecutive waypoints. The magnitudes of these vectors represent the desired speed and are sent to the longitudinal controller to generate $throttle$ and $brake$ control actions, and the orientations are sent to the lateral controller to get the $steer$ action.

### 3.2.2 Multi-step control prediction branch

As discussed in Sec. 3.1, for a control model predicting current control actions based on current input only, the supervised training is just behavior cloning, which relies on the independent and identically distributed (IID) assumption. This assumption apparently does not hold because of the distribution shifts in test cases, since the closed-loop tests require sequential decision making where the historical actions will affect the future states and actions. Instead of modeling it as a Markov Decision Process (MDP) and resorting to reinforcement learning, here we devise a simple way to mitigate the problem by predicting multi-step control into the future.

Given the current state $\mathbf{x}_t$, now our multi-step control prediction branch outputs multiple actions: $\pi_{\theta_{multi}} = (\mathbf{a}_t, \mathbf{a}_{t+1}, \cdots, \mathbf{a}_{t+K})$. However, it is difficult to predict future control actions since we only have sensor inputs at the current time step. Towards this problem, we devise a temporal module to implicitly carry out the changing and interaction process of the environment and our agent. It is supposed to provide mainly dynamic information about the environment and the status of the agent

itself, such as the motion of other objects, the changing of traffic lights, and the status of the ego agent. Meanwhile, to improve the ability of incorporating critical static information (*e.g.*, curbs and lanes) and boost the spatial consistency of two branches, we propose to use the trajectory branch to guide the control counterpart to attend to proper regions of the input image at each future time step.

**Temporal module.** Our temporal module is implemented with a GRU for better consistency with the trajectory branch. At step $t$ $(0 \leq t \leq K - 1)$, the input for the temporal module is the concatenation of the current feature $\mathbf{j}_t^{\text{ctl}}$ (more construction details in the next section) and current predicted action $\mathbf{a}_t$, which is a compact representation about the current states of the environment and the agent itself. The temporal module is supposed to reason about the dynamic changing process based on current feature vector and the predicted action. Then the updated hidden state $\mathbf{h}_{t+1}^{\text{ctl}}$ will contain dynamic information about the environment and the updated status of the agent at time step $t + 1$. To some extent, the temporal module acts as a coarse simulator with the whole environment and the agent being abstracted as a feature vector. It then simulates the interaction between the environment and the agent based on current prediction of actions.

**Trajectory-guided attention.** With the sensor input at current step only, it is hard to pick out desirable regions where the model should focus on at future steps. However, the location of the ego agent contains important cues about how to find those regions containing critical static information for control prediction at each step.

Therefore, we seek help from the trajectory planning branch to get information about the possible location of our agent at that corresponding step. As shown in Fig. 3, TCP implements this by learning an attention map to extract important information from the encoded feature map. The interaction between two branches enhances the consistency of these two strongly related output paradigms and further elaborates the multi-task spirit. Specifically, with the 2D feature map extracted by the image encoder $\mathbf{F}$ at time step $t$ $(1 \leq t \leq K)$, we calculate an attention map $\mathbf{w}_t \in \mathbb{R}^{1 \times \text{H} \times \text{W}}$ using the corresponding hidden states from the control branch and the trajectory branch:

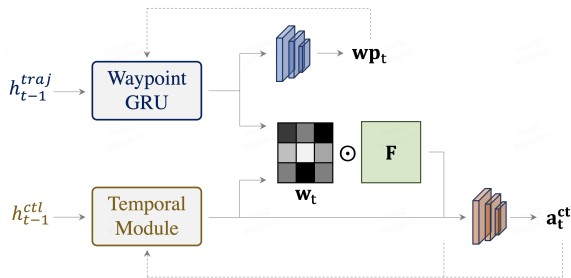

Figure 3: Detailed trajectory guiding process. For predictions at time step $t$, the hidden states from the waypoint GRU and the temporal module are combined to learn an attention weight map to re-aggregate the 2D image feature map for control prediction.

$$\mathbf{w}_t = \text{MLP}(\text{Concat}[\mathbf{h}_t^{\text{traj}}, \mathbf{h}_t^{\text{ctl}}]). \tag{2}$$

The attention map $\mathbf{w}_t \in \mathbb{R}^{1 \times \text{H} \times \text{W}}$ is adopted to aggregate the feature map $\mathbf{F}$ for this step. We then combine the attended feature map with $\mathbf{h}_t^{\text{ctl}}$ to form the informative representation feature $\mathbf{j}_t^{\text{ctl}}$ containing both static and dynamic information about the environment and the ego agent. The process can be described as follows:

$$\mathbf{j}_t^{\text{ctl}} = \text{MLP}(\text{Concat}[\text{Sum}(\text{Softmax}(\mathbf{w}_t) \odot \mathbf{F}), \mathbf{h}_t^{\text{ctl}}]). \tag{3}$$

The informative representation feature $\mathbf{j}_t^{\text{ctl}}$ is fed into a policy head which is shared among all time steps to predict the corresponding control action $\mathbf{a}_t$. Note that for the initial step, we only use the measurement feature to calculate the initial attention map and combine the attended image feature with the measurement feature to form the initial feature vector $\mathbf{j}_0^{\text{ctl}}$. To guarantee the feature $\mathbf{j}_t^{\text{ctl}}$ does describe the state at that step and contain the important information for control prediction, we add a feature loss at each step to make $\mathbf{j}_t^{\text{ctl}}$ close to the feature of the expert as well.

To this end, our TCP framework endows the model with the reasoning ability along a short time horizon. It emphasizes how to make current control prediction close to the one from the expert. Furthermore, it takes into account what current control prediction can make the environment states and status of ego agent in future time steps similar to the ones from the expert.

### 3.3 Loss Design

Our loss contains trajectory planning loss $\mathcal{L}_{traj}$, control prediction $\mathcal{L}_{ctl}$, and auxiliary loss $\mathcal{L}_{aux}$.

For the trajectory planning branch, the loss $\mathcal{L}_{traj}$ can be expressed as:

$$\mathcal{L}_{traj} = \sum_{t=1}^{K} \|\mathbf{wp}_t - \mathbf{\hat{wp}}_t\|_1 + \lambda_F \cdot \mathcal{L}_F \left(\mathbf{j}_0^{\text{traj}}, \mathbf{j}_0^{\text{Expert}}\right), \tag{4}$$

where $\mathbf{wp}_t$, $\mathbf{\hat{wp}}_t$ are the predicted and ground truth waypoint at the $t^{th}$ step respectively. $\mathcal{L}_F$ indicates the feature loss measuring the $L_2$ distance between $\mathbf{j}_0^{\text{traj}}$ and the feature $\mathbf{j}_0^{\text{Expert}}$ from the expert at the current step as an additional supervision signal [60]. $\lambda_F$ is a tunable loss weight.

For the control prediction branch, we model the action as a beta distribution. The loss $\mathcal{L}_{ctl}$ is:

$$\begin{aligned}
\mathcal{L}_{ctl} =& \mathbf{KL}\left(\text{Beta}(\mathbf{a}_0)||\text{Beta}(\mathbf{\hat{a}}_0)\right) + \frac{1}{K}\sum_{t=1}^{K} \mathbf{KL}\left(\text{Beta}(\mathbf{a}_t)||\text{Beta}(\mathbf{\hat{a}}_t)\right) \\
&+ \lambda_F \cdot \mathcal{L}_F\left(\mathbf{j}_0^{\text{ctl}}, \mathbf{j}_0^{\text{Expert}}\right) + \frac{1}{K}\sum_{t=1}^{K} \mathcal{L}_F\left(\mathbf{j}_t^{\text{ctl}}, \mathbf{j}_t^{\text{Expert}}\right),
\end{aligned} \tag{5}$$

where $\text{Beta}(\mathbf{a})$ denotes the beta distribution represented by the corresponding predicted distribution parameters and KL-divergence is used to measure the similarity between the predicted control distribution and the one from expert, *i.e.*, $\text{Beta}(\mathbf{\hat{a}})$. Feature loss is applied here as well. Note that all losses for future time steps ($t \geq 1$) are averaged and then added to the loss for the current time step ($t = 0$), since the action executed immediately should be our key target to optimize.

To help the agent better estimate its current state, we add a speed prediction head to predict current speed $s$ from the image feature and a value prediction head to predict the expected return estimated by the expert, similarly as in [60]. We take the $L_1$ loss for the speed prediction and $L_2$ loss for the value prediction, denoting their weighted sum as $\mathcal{L}_{aux}$.

The overall loss is as follows, as weighted by $\lambda_{traj}, \lambda_{ctl}, \lambda_{aux}$:

$$\mathcal{L} = \lambda_{traj} \cdot \mathcal{L}_{traj} + \lambda_{ctl} \cdot \mathcal{L}_{ctl} + \lambda_{aux} \cdot \mathcal{L}_{aux}. \tag{6}$$

## 3.4 Output Fusion

We have two forms of output representations from our TCP framework: the planned trajectory and the predicted control. To further combine their advantages, we devise a situation-based fusion strategy as depicted in Algorithm 1. Specifically, denote $\alpha$ as a combination weight whose value is between 0 to 0.5, in a certain situation where one representation is more suitable according to our prior belief, we combine the results from trajectory and control predictions by taking average with weight $\alpha$ so that the more suitable one takes up more weight $(1 - \alpha)$. Note that the combination weight $\alpha$ indeed does not need to be

---

**Algorithm 1:** Situation based fusion scheme to combine the two output paradigms

**Input:** sensory input $\mathbf{i}$, speed of the ego vehicle $v$, high level navigation information $\mathbf{g}$.
**Hyper parameters:** combination weight $\alpha \in [0, 0.5]$
**Output:** final control signals $\mathbf{a}$

$\{\mathbf{wp}_t\}_{t=0}^{K}, \mathbf{a}^{\text{ctl}} \leftarrow \text{TCP}(\mathbf{i}, v, \mathbf{g})$
$\mathbf{a}^{\text{traj}} \leftarrow \text{Low-level Controller}(\{\mathbf{wp}_t\}_{t=0}^{K})$
Get current *situation*
**if** *situation* is *trajectory specialized* **then**
    $\mathbf{a} \leftarrow \alpha \times \mathbf{a}^{\text{ctl}} + (1-\alpha) \times \mathbf{a}^{\text{traj}}$
**else**
    $\mathbf{a} \leftarrow \alpha \times \mathbf{a}^{\text{traj}} + (1-\alpha) \times \mathbf{a}^{\text{ctl}}$
**end**

---

a constant or symmetric, which means we can set it to different values under different situations or different for specific control signals. In our experiment, we choose the *situation* according to whether the ego vehicle is turning, implying that if it is turning, the *situation* is *control specialized* otherwise *trajectory specialized*.

# 4 Experiments

## 4.1 Experimental Setup

**Task & Evaluation metrics.** Our method is validated and tested in the CARLA driving simulator [20]. Given a route defined by a sequence of sparse navigation points together with high level commands (straight, turn left/right, lane changing, and lane following), the closed-loop driving task

Table 1: Evaluation on the public CARLA Leaderboard [1] (accessed in May 2022). Our method TCP and TCP-Ens achieve a driving score of 69.714 and 75.137 respectively with only a monocular camera. More detailed infraction statistics can be found in the Supplementary.

| Rank | Method | Sensor Inputs | | Key Metrics ↑ | | |
|------|--------|---------------|-------|---------------|-----------------|------------------|
| | | #Cameras | LiDAR | Driving Score | Route Completion | Infraction Score |
| 1 | **TCP-Ens** (ours) | 1 | ✗ | **75.137** | 85.629 | **0.873** |
| 1 | **TCP** (ours) | 1 | ✗ | **69.714** | 82.962 | **0.851** |
| 1 | **TCP-SB** (ours) | 1 | ✗ | **68.695** | 82.957 | **0.833** |
| 2 | LAV [10] | 4 | ✓ | 61.846 | **94.459** | 0.640 |
| 3 | Transfuser | 3 | ✓ | 61.181 | 86.694 | 0.714 |
| 4 | Latent Transfuser | 3 | ✗ | 45.029 | 75.366 | 0.618 |
| 5 | GRIAD [9] | 3 | ✗ | 36.787 | 61.855 | 0.597 |
| 6 | Transfuser+ [29] | 4 | ✓ | 34.577 | 69.841 | 0.562 |
| 7 | WoR [12] | 4 | ✗ | 31.370 | 57.647 | 0.557 |
| 8 | MaRLn [50] | 1 | ✗ | 24.980 | 46.968 | 0.518 |
| 9 | NEAT [15] | 3 | ✗ | 21.832 | 41.707 | 0.650 |

Table 2: Comparison between the control and trajectory only model in terms of infractions frequency. TurnRatio means the corresponding ratio of happening during turning.

| Model | Driving Score | Collisions vehicles | | Collisions layout | | Off-road infractions | | Agent blocked | |
|-------|---------------|---------|-----------|---------|-----------|---------|-----------|---------|-----------|
| | | #/km ↓ | TurnRatio | #/km ↓ | TurnRatio | #/km ↓ | TurnRatio | #/km ↓ | TurnRatio |
| Control-Only | 32.45±2.23 | 1.25 | 50.90% | **0.23** | 10.00% | **0.59** | 46.15% | **0.41** | 50.00% |
| Trajectory-Only | 28.29±3.03 | **0.85** | 38.70% | 0.77 | 64.20% | 0.74 | 62.90% | 0.77 | 64.20% |

requires the autonomous agent to drive towards the destination point. It is designed to simulate realistic traffic situations and includes different challenging scenarios such as obstacle avoidance, crossing an unsignalized intersection, and sudden control loss. There are three major metrics: Driving Score, Route Completion, and Infraction Score. Route Completion is the percentage of the route completed by the autonomous agent. Infraction Score measures the number of infractions made along the route, with pedestrians, vehicles, road layouts, red lights, and *etc*. Driving Score is the main metric which is the product of Route Completion and Infraction Score.

**Dataset.** We use randomly generated routes under random weather conditions to collect 420K data in the 8 public towns offered by the CARLA simulator. Similar to [10], we train TCP on 189K of data in 4 out of 8 towns (Town01, Town03, Town04, and Town06) for ablations and train with all 420K data for our online leaderboard submission.

## 4.2 State-of-the-art Comparison

Table 1 shows the result of the comparison between our method and the top 8 entries on the public CARLA Leaderboard [1]. We report the results of TCP and two variants. **TCP-SB** replaces shared encoders of TCP with two separate ones for two branches, and **TCP-Ens** is the ensemble of TCP and TCP-SB. Our method TCP-Ens ranks first on the leaderboard with a 75.137 driving score and highest infraction score, and TCP alone also surpasses prior methods. Note that our method only uses a monocular camera while the top 2-4 methods all use multiple cameras and a LiDAR. Our driving score is 50.157 higher than the second-best monocular camera method, MaRLn [50]. Our route completion is slightly inferior to the LiDAR candidates - one reason is that methods using LiDAR may have a better object detection ability. Based on the detection results, they usually adopt a crawling strategy, indicating that the vehicle would move slowly when it has stopped for a long time and there are no obstacles ahead. As described in [29], this could alleviate ego vehicle's blocking problems to boost the route completion performance.

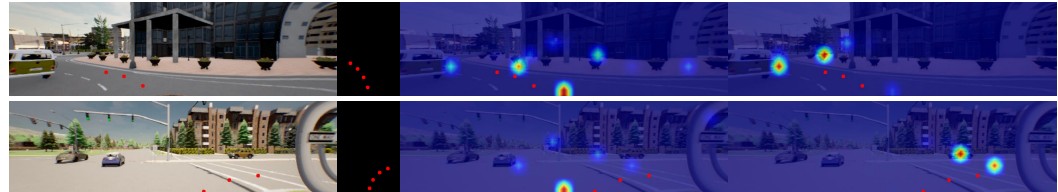

Figure 4: The trajectory-guided attention maps in two cases. In each case (row), from the left to right we show that the input image with the predicted trajectory (the first waypoint is projected out of the image), the predicted trajectory in the top-down view, the attention map $\mathbf{w}_1$, the attention map $\mathbf{w}_3$.

Table 3: Ablative study on the effectiveness of different components design of our model.

| Exp. | Driving Score | Route Completion | Infraction Score |
|---|---|---|---|
| Control | 32.45±2.23 | 76.54±3.22 | 0.45±0.03 |
| + traj-task | 34.98±1.96 | 81.32±5.50 | 0.49±0.05 |
| + temporal | 42.87±4.77 | **87.51**±3.63 | 0.49±0.07 |
| + traj-attn | 46.08±3.47 | 84.95±1.84 | 0.56±0.03 |
| + fusion | **57.01**±1.88 | 85.27±1.20 | **0.67**±0.01 |

Table 4: Comparison between MTL and ensemble methods ($\alpha$ is 0.3 for all experiments).

| Exp. | Driving Score | #Param. | FLOPs | FPS |
|---|---|---|---|---|
| Ensemble | 45.03±1.28 | 46.81M | 17.07G | 69.47 |
| MTL | 48.27±0.58 | 23.58M | 8.54G | 133.30 |
| TCP-SB | 52.46±4.66 | 47.26M | 17.07G | 69.35 |
| TCP | 57.01±1.88 | 25.77M | 8.54G | 125.71 |
| TCP-Ens | 59.09±3.66 | 73.03M | 25.61G | 44.70 |

### 4.3 Control vs. Trajectory

In this section, we conduct quantitative experiments to compare the **Control-Only** model and the **Trajectory-Only** model to demonstrate their advantages and disadvantages. For both models, we use the same setting except for the output head and its corresponding loss. We use a ResNet-34 to encode visual inputs and a measurement module to encode the navigation information. Similar to [60], we add speed and value heads as auxiliary tasks to help the model better encode the environment. For **Control-Only**, we predict the control distribution based on the concatenated latent feature from the two encoders. As for **Trajectory-Only**, we feed the feature to a GRU decoder to generate waypoints. As shown in Table 2, though Trajectory-Only collides with vehicles less frequently than Control-Only, it has more layout collisions, off-road infractions, and agent blocks. We also count the ratio of each kind of infraction that occurs during turning. It can be observed that for Trajectory-Only, a large portion of such infractions happen when the ego agent is turning compared to Control-Only. This has verified that Trajectory-Only performs worse when the agent is turning, which is probably caused by the unsatisfactory trajectory following performance of simple PID controllers as discussed in Sec. 1. As for the fact that Control-Only has a higher vehicle-collision rate, it is because the model focuses on the current time step and the reaction to potential collisions tends to be late, as depicted in Sec. 1 as well. The results above further validate the necessity of combining the two output paradigms.

### 4.4 Ablative Study and Visualization

**Component analysis.** We first validate the effectiveness of the trajectory-guided multi-step control prediction design, as shown in Table 3. We only employ the control branch output except for the last complete one when fusion is applied for these ablations. Adding a trajectory branch as an auxiliary task improves the performance by 2.5 points. The multi-step predictions with our temporal module greatly help with 7.9 points gain, and adding the trajectory-guided attention further acquires an improvement of 3.2 points. Finally, applying our situation based fusion scheme ($\alpha$ is set to 0.3) significantly boosts the infraction score, leading the overall driving score to 57.

**Multi-task vs. Ensemble.** The comparison regarding their performances and computational complexity is given in Table 4. **Ensemble** denotes directly combining the outputs of Control-Only and Trajectory-Only with our situation based fusion scheme. **MTL** represents the model with a shared CNN backbone and measurement encoders followed by a trajectory branch and a control branch, but the control branch predicts current step prediction only and there are no interactions between the two branches. We conclude that directly combining two models with our fusion scheme greatly improves the performance, and using an MTL approach works better than ensemble but with a much

smaller model size and GFLOPs. A conventional ensemble approach to combine results from TCP and TCP-SB as TCP-Ens brings further performance gain at the cost of computational complexity.

**Situation based fusion weight.** We investigate the choice of the combination weight $\alpha$ in the situation based fusion scheme and show the box plot of the driving scores in the figure to the right. Besides $\alpha \in [0, 0.5]$, we additionally test 0.7 and 1, meaning that two results are conversely mixed with our *specialization* definition. We see that only using the control from the specialized branch $\alpha = 0$ performs poorly while directly taking the average or fusing conversely still has comparable results. One reason is that the *situation* criterion used here is whether the vehicle is turning, making most cases *trajectory specialized*, and the stronger control branch is not utilized enough if $\alpha$ is small. Note that the situation based fusion scheme is general and flexible, and the criterion or $\alpha$ value used here is relatively coarse.

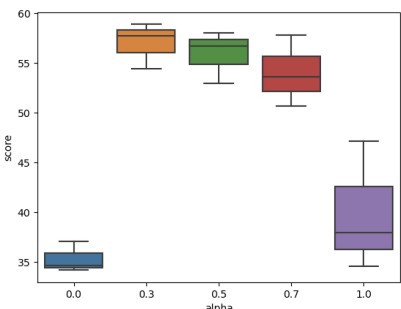

Figure 5: Box plot of the driving score with different $\alpha$ values (3 trials for each $\alpha$).

**Visualization.** Fig. 4 visualizes the trajectory-guided attention maps. The trajectory branch provides location-related information to guide the control branch to focus on important regions which are useful for future control prediction. See more qualitative results in the Supplementary.

## 5 Conclusion

In this work, we study two learning and prediction paradigms based on trajectory and direct control, respectively, for end-to-end autonomous driving. We propose a unified framework comprised of a trajectory branch and a novel multi-step control branch with interactions in between. We design a situation based fusion scheme to combine the results from two branches. Our method with only a monocular camera has achieved state-of-the-art performance on the CARLA Leaderboard.

## Acknowledgments

This work was partly supported by National Key Research and Development Program of China (2020AAA0107600), NSFC (62206172, 61972250), Shanghai Municipal Science and Technology Major Project (2021SHZDZX0102), and Shanghai Committee of Science and Technology (21DZ1100100, 22511105100).

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
