# OpenReview forum: "Trajectory-guided Control Prediction for End-to-end Autonomous Driving: A Simple yet Strong Baseline"
_NeurIPS.cc/2022/Conference — NeurIPS 2022 Accept_

### Official Review · Reviewer_ZcJ9 · 2022-07-11

**Rating:** 7
**Confidence:** 4
**Soundness:** 3 good
**Presentation:** 3 good
**Contribution:** 3 good

**Summary:**

This paper presents TCP, a novel trajectory predicting/controlling network architecture for vision-based end-to-end autonomous driving. At test time, TCP ensembles the control outputs from a direct prediction branch and a trajectory prediction  w/ PID controllers. Both branches use GRU to predict waypoints/controls auto-regressively and are jointly trained. At each timestep, the control prediction branch additionally takes as input attention-weighted image backbone features computed from the trajectory prediction branch.


**Questions:**

Apart from the ablation mentioned in the Strengths & Weaknesses section, I would like to hear from the authors how they think TCP will work in a real world setting, what might be the main obstacles and bottlenecks that the proposed method could have when scaling up to real world data.


**Limitations:**

Same as my question in the section above – I would like to hear from the authors what they think might be the limitations of the proposed method when scaling to the real world.


**Strengths And Weaknesses:**

=== Strengths ===
+ The strongest aspect of this paper is its impressive performance on the challenging CARLA leaderboard. The fact that TCP is capable of attaining such a strong performance with a simple vision backbone is inspiring, suggesting prior methods for end-to-end driving in CARLA might all have suboptimal controller designs.
+ The ablation studies are mostly comprehensive, and the design choices are well justified.
+ The attention map visualization is insightful and shows what areas are attended for the control prediction task.

=== Weaknesses ===
- I do not have any major complaints regarding the technical content. My main question is how much performance gain comes from fusing the trajectory/control prediction branches as the title and paper implies, or suboptimal PID parameters for the trajectory-only branch, and the trajectory-attended image features inputs in the autoregressive prediction stage. I would like to see a trajectory-only variant where, similar to the proposed situation-based fusion, has situation-based PID controller parameters (different PID gains based on different high-level commands, or whether it’s turning etc.), which still has the novel trajectory-computed attention fusion module (i.e. the attention-weighed image features get consumed by the trajectory branch instead of the control branch). This will further help distinguish the main source of improvements among the design choices.

---

> ### Author Response · Authors · 2022-08-02
> **Author response to Reviewer Zcj9**
>
> Dear Reviewer Zcj9,
>
> Thank you for appreciating our work. We address Reviewer's concerns below.
>
> **Q1: Weakness. Trajectory-Only variant to have situation-based PID controller for the Trajectory-Only model.**
>
> **A1:** Thanks for the suggestion. In this rebuttal, we implement the `Trajectory-Only` model with `Trajectory-attended` image features and test them both with original `PID` and `Situation-based PID`.  The experiment results are listed below.
>
> |  | Driving Score | Route Completion | Infraction Penalty | Collisions vehicles | Collisions layout | Off-road infractions | Agent blocked | Red light infractions |
> |:------------------------------------------:|:-------------------:|:-----------------:|:--------------------:|:-------------------:|:-----------------:|:--------------------:|:-------------:|:---------------------:|
> |Trajectory-Only (Original PID)|     28.29     |       58.11      |        0.50        |         0.85        |0.77 |0.74| 0.77|0.41|
> |   Trajectory-Only (Situation-based PID )   |     30.63     |       66.13      |        0.53        |0.68|0.32 |         0.54         |      0.54     |          0.59         |
> |     Trajectory-attended (Original PID)     |     26.35     |       70.64      |        0.38        |         1.70        |        0.23       |         0.47         |      0.49     |          0.23         |
> | Trajectory-attended (Situation-based PID ) |     30.84     |       76.09      |        0.41        |         1.78        |        0.16       |         0.38         |      0.40     |          0.29         |
>
>
> >  Experiment settings: we have tuned parameters of our PID module in some routes, as provided in the official CARLA leaderboard; these routes are from different towns compared with the ones for our validation evaluation. Specifically, two sets of PID parameters are generated so that we can alter according to current situation (whether the vehicle is turning in this case) similar to the proposed fusion mechanism.
>
> Although we tune the parameters carefully and obtain two sets of parameters to choose, **the performance improvement is trivial** (see Row `1 vs 2`, `3 vs 4` respectively).
> This is because the environment for parameter tuning is different from that for evaluation (such as the road topology and the curvature of turnings).
> This actually **validates our motivation to avoid onerous parameter tuning**. Besides, the heuristic tuning may still perform poorly in new environment.
> On the elaboration of `Trajectory-attended vs Trajectory-Only`,
> the original trajectory-guided attention is designed to guide the control prediction at each future step, and we treat all four waypoints together as the current policy for current step. Using its previous waypoint prediction to re-aggregate the image and combining it with the hidden state in the GRU of the trajectory branch make the training and waypoints prediction process less stable.
>
>
> **Q2: Questions and Limitations: whether TCP will work when scaling to the real world; what might be the main obstacles and bottlenecks.**
>
> **A2:** Thank you for pointing out the potential limitations of TCP. We have added discussion about possible problems when scaling our model to real world in the **Limitation** section of the revised Supplementary (Sec. D.1.2). There are several aspects to consider for real world setting:
>
>
> - **Balance Data distribution.** The real world driving scenario is more complex and diverse than the simulation, and the data distribution is highly skewed. We need to carefully adjust the training data manually to assure different driving maneuvers are distributed more uniformly. And we need to assure those rare but safety-critical cases can be captured by our model.
> - **Ensure the generalizability of the perception model.** Since the scene looks very different due to weather and time changes in real world, it is important for our perception model to learn a robust and generalizable representation of the scene which is invariant to irrelevant factors. The possible approach is to add auxiliary tasks like depth estimation and semantic segmentation.
> - **Incorporate motion prediction and object detection modules.** To improve safety and explainability, additional modules to detect other objects and make predictions about other dynamic agents could be combined with our model.
> - **Learn from corrections.** Only learning from perfect human demonstrations may make the model fail during testing due to the distribution shift problem. It is important to utilize techniques like DAgger to let the model learn corrections in an online setting.
> - **Improve open-loop evaluation for real world.** It is natural to test the model in closed-loop in simulation, but the cost for real world closed-loop evaluation is too high. Therefore, we need to improve the evaluation approaches based on offline dataset to get a good indicator of the performance in real world setting.

---

> > ### Comment · Reviewer_ZcJ9 · 2022-08-07
> > **Thanks for the response**
> >
> > Thanks the authors for their detailed response. It is interesting to see the trajectory-attended model attains significantly higher route completion despite having worse infractions. Can the authors further provide detailed infraction comparison between the two variants? e.g. what kind of infractions do the trajectory attended model have more compared to the original one. I would also like to thank the authors for adding the limitations discussion of the paper.

---

> > > ### Author Response · Authors · 2022-08-08
> > > **Author Response to Reviewer Zcj9**
> > >
> > > Thanks for the follow-up discussion. We have updated detailed infraction statistics in the table above in **A1**. We can observe that trajectory-attended variants have higher *Collisions vehicles* rates, while other penalties (e.g., *Collisions layout* and *Off-road infractions* rates) are all lower compared with the original Trajectory-Only models. We hypothesize that using future waypoint for the attention weight calculation makes the model focus on those location-related and static information (e.g., curbs and lanes) more. This also validates our motivation to utilize trajectory waypoints to guide the multi-step control prediction.

---

### Official Review · Reviewer_cv3D · 2022-07-11

**Rating:** 3
**Confidence:** 5
**Soundness:** 2 fair
**Presentation:** 2 fair
**Contribution:** 2 fair

**Summary:**

This work presents a novel approach (TCP) that combines trajectory planning and control prediction in a multitask learning framework for end-to-end autonomous driving. It analyzes the limitations of both paradigms (Fig. 1), eg. inertial problem, incorrect turnings, and single-step prediction, and proposes a multi-step control prediction module with trajectory-guided attention and situation-based fusion scheme to incorporate the best of both worlds. Extensive experiments on the CARLA leaderboard (Table 1) show state-of-the-art performance by a wide margin, with only a monocular camera, compared to baselines that use multiple sensors as input. Moreover, the ablation study (Table 2,3,4) and visualizations (Fig. 4 and supplementary) provide valuable insights into the capabilities of the proposed approach.

**Questions:**

Based on the weaknesses mentioned above, the following experiments are required to verify the claims in the paper:
- Trajectory-only model with feature loss applied to each prediction timestep and auxiliary loss (on speed and value prediction) used as well. This requires training a new model and should be compared in Table 3.
- A single GRU module to predict both future waypoints and multi-timestep control predictions from the hidden state of the GRU. This requires training a new model and should be compared in Table 3.
- Direct ensemble (average control predictions) of control-only and trajectory-only models in Table 4. This does not require training any model.
- An ensemble of multi-step control prediction and trajectory prediction (both situation-based fusion and direct averaging should be considered) without the trajectory-guided attention module. This does not require training any model and should be compared in Table 4.

Certain parts need clarification:
- L194-195: How does the multi-step control prediction branch mitigate the IID assumption?
- L202-204: How does the trajectory branch improve the ability to incorporate static layout information (eg. curbs)?
- L309-L313: Why does TCP has lower route completion compared to LiDAR-based methods. Is it because the TCP gets blocked more often than other methods? It is also possible that taking the maximum of brake values from different models for the ensemble (L33-34 in supplementary) leads to an overly conservative policy.
- The evaluation protocol for the experiments in Table 2,3,4 is different than Table 1 which uses the CARLA leaderboard. The authors should include a brief description of this evaluation protocol in the paper.
 - Fig. 1(a) shows a failure case at turnings for trajectory-based methods. Since the details of the PID controllers are not provided in the paper, it is unclear if the PID controller could just be tuned to better follow the trajectory.
 - Implementation details regarding the GRU modules, MLPs in the trajectory-guided attention, and weights for loss terms (Eq. 4,5,6) should also be included in the paper.

Additional suggestions to improve the paper:
- As mentioned earlier, it would be helpful if the authors could also provide comparisons to FASNet. This would greatly improve the paper.
- It would be interesting to visualize the GradCAM (Selvaraju et al. IJCV 2019) attention maps for the multi-step control prediction branch without the trajectory-guided attention module. This could provide useful insights into whether the multi-step control prediction helps the model to focus on important regions for future control predictions. These GradCAM attention maps can be compared to attention maps in Fig. 4 to verify the utility of trajectory-guided attention.

**Limitations:**

The authors have provided a discussion on limitations and societal impact in the supplementary. Additional suggestions are mentioned above.

### Update after Rebuttal ###
I appreciate additional ablations and discussions with authors which helped me get a better understanding of the paper. After rebuttal and discussion with other reviewers, I am retaining my original score.

My main concern is that majority of the gain is coming from the heuristic $\alpha$ used for ensembling. Table 2 in the main paper shows Control-only=`32.45`, trajectory-only=`28.29`. In one of the ablations provided by the authors in the rebuttal, ensemble of control-only & trajectory-only methods with non-constant $\alpha$ gives a score of `50.98`. TCP has a score of `57.01` in Table 4 in the main paper. The ensemble heuristic leads to a gain of around `19` points whereas all the architectural modifications - shared backbone, multi-step control, temporal module, trajectory-guided attention - combined result in a gain of `6` points.

While it might be possible to use the fusion mechanism in a general and flexible way (eg. as a mixture of experts or using probabilistic uncertainty, as mentioned by the authors in the rebuttal), the paper currently uses a heuristic for fusion. Based on this, I agree with reviewer `6yad` that the paper does not have sufficient technical contribution for NeurIPS, and venues like IROS, ICRA are a better fit. The authors have mentioned multiple interesting directions, eg. link between the temporal module and world models, using mixture of experts or probabilistic uncertainty for fusion. I encourage the authors to explore these directions.

**Strengths And Weaknesses:**

Strengths:

- The idea to combine trajectory planning and control prediction in a multitask learning framework is simple and easy to understand. Algorithm 1 and Fig. 2, 3 are great and provide a clear understanding of the proposed approach.
- TCP achieves state-of-the-art results on the official CARLA leaderboard, surpassing the previous best model by 13.29 points. This is quite impressive since TCP uses a single monocular camera as input whereas the other top models use multiple sensor inputs: multiview camera and LiDAR.

Weaknesses:

- The loss function for the control branch (Eq. 5) contains feature loss for the entire prediction horizon whereas that is not the case for the trajectory branch (Eq. 4). Since the trajectory branch contains a GRU that outputs a hidden state at each prediction timestep, the feature loss (from t=1 to K) could also be applied in Eq. 4 at each future timestep. The authors should include an ablation with a trajectory-only model with feature loss applied to each prediction timestep and auxiliary loss (on speed and value prediction) used as well.

- What is the reason for using 2 separate GRU branches for predicting waypoints and control values? Since both waypoints and control are alternate representations of driving behavior, they could be directly predicted from a single branch. Since the waypoint branch also contains a GRU, the hidden state of that GRU could be used to predict both waypoints and control values through separate MLP heads. It is not clear why a separate temporal module is required since it is already present in the trajectory branch.

- The authors should compare to a baseline which does a direct ensemble (average control predictions) of control-only and trajectory-only models in Table 4. This is important to understand if the situation-based fusion scheme is indeed better than simple averaging.

- Does the Control-Only model in Table 2 and L340 predict only single-step control values? If yes, the authors should also compare to a baseline that does an ensemble of multi-step control prediction and trajectory prediction (both situation-based fusion and direct averaging should be considered) without the trajectory-guided attention module.

- L121-122 mentions that the proposed idea is closely related to FASNet. It would be helpful if the authors could also provide comparisons to FASNet. Since the code of FASNet is not publicly available, implementing, training and testing may not be possible in time for rebuttal but this would greatly improve the paper.

- Sec. B in supplementary mentions that for online submission to the CARLA leaderboard, $\alpha=0.5$ is used if situation is trajectory specialized whereas $\alpha=0$ is used when it is control specialized. Also, the maximum of brake is used instead of taking the average. It'd be great if the authors could explain the reasoning behind using a different fusion scheme. Also, how is $\alpha$ selected? Is there a separate validation protocol for tuning these values?

- Relevant references that should also be included:
    - Zhihao Li, Toshiyuki Motoyoshi, Kazuma Sasaki, Tetsuya Ogata, Shigeki Sugano. Rethinking self-driving: Multi-task knowledge for better generalization and accident explanation ability. arXiv 1809.11100, 2018.
    - Alex Kendall, Jeffrey Hawke, David Janz, Przemyslaw Mazur, Daniele Reda, John-Mark Allen, Vinh-Dieu Lam, Alex Bewley, Amar Shah. Learning to Drive in a Day. ICRA 2019
    - Jeffrey Hawke, Richard Shen, Corina Gurau, Siddharth Sharma, Daniele Reda, Nikolay Nikolov, Przemyslaw Mazur, Sean Micklethwaite, Nicolas Griffiths, Amar Shah, Alex Kendall. Urban Driving with Conditional Imitation Learning. ICRA 2020
    - Nicholas Rhinehart, Rowan McAllister, Sergey Levine. Deep Imitative Models for Flexible Inference, Planning, and Control. ICLR 2020
    - Albert Zhao, Tong He, Yitao Liang, Haibin Huang, Guy Van den Broeck, Stefano Soatto. SAM: Squeeze-and-Mimic Networks for Conditional Visual Driving Policy Learning. CoRL 2020
    - Yi Xiao, Felipe Codevilla, Akhil Gurram, Onay Urfalioglu, Antonio M. López. Multimodal End-to-End Autonomous Driving. arXiv 1906.03199, 2019

---

> ### Author Response · Authors · 2022-08-02
> **Author response to Reviewer cv3D (1/2: New Experiments)**
>
> Dear Reviewer cv3D,
>
> Thanks for the detailed and helpful review; we really appreciate it. The main concerns in the review feedback concentrate on technical clarifications and additional experiments to verify the claims. We address each comment in details below.
>
> **Q1: Difference between two GRUs in control and trajectory branches.**
>
> > This corresponds to requested ablations on 1) trajectory-Only model with feature loss applied to each prediction timestep, 2) a single GRU module to predict both future waypoints and multi-timestep control predictions.
>
> **A1:** Thanks for the comment. We explain it from two perspectives below.
>
> **Technical clarification.** We might argue that the two GRUs have **different roles**. For the trajectory branch, the policy output for **current step** is a series of waypoints (4 waypoints all together), without involving planning at future steps. The GRU in the trajectory branch is the policy head, and it can be replaced by other non-recursive implementations such as MLPs to output waypoints all at once. The GRU in the control branch works as the **temporal module**, aiming to encode latent dynamics for the future step based on current feature and action. The control branch makes multiple policy outputs for future steps with the temporal module. Our temporal module aims to match the policy-related future features with those from the expert. The temporal module cannot generate future states all at once.
>
> **Additional ablations.** Since the GRU in the trajectory branch is just the policy head for current step, it's not reasonable to directly add future feature supervision at each step. Therefore, we modify the role of it from a policy head to the "temporal module", and the policy head is now implemented using an MLP. Specifically, the trajectory branch only regresses one waypoint as the policy output for each step. The modified GRU takes in current feature and the single waypoint, and generates latent feature for the next step, based on which the policy head (MLP) regresses the next waypoint. In this way, the future feature loss could be added to each step. We refer this model as **`Traj-Only-multistep`**. Another policy head for control (implemented by another MLP) is added to Traj-Only-multistep so that we get both the waypoint and the control action with **a single GRU**, and we call it **`MTL-2heads`**. The performance of these variants is listed below.
>
> ||Driving Score|Route Completion|Infraction Penalty|
> |:-------------------------:|:-----:|:-----:|:-----:|
> |Trajectory-Only (Original)| 28.29 |58.11|0.50|
> |Traj-Only-multistep|26.30|60.78|0.46|
> |MTL (2nd row in Table 4)|48.27|81.62|0.60|
> |MTL-2heads|43.42|79.51|0.56|
>
> Traj-Only-multistep performs slightly worse than the original Trajectory-Only model. This is probably because the policy head now predicts one waypoint only, making it less stable. Combining two branches with a shared GRU harms the performance. Though we combine these two branches in an MTL approach in the original TCP model, these two tasks still have intrinsic differences. Therefore, using different MLP layers at the last stage alone to generate different outputs could hinder performance.
>
> **Q2: Does the Control-Only model predict only single-step control values. If yes, compare to new experiments of direct ensemble (simple average), an ensemble of multi-step control prediction and trajectory prediction without trajectory-guided attention.**
>
>
> **A2:**
>
> ||Driving Score|Route Completion|Infraction Penalty|
> |:-----------------------------------------:|:-----:|:-----:|:-----:|
> |Ensemble ($\alpha$ = 0.3, 1st row in Table4)|45.03| 79.30 |0.59|
> |Ensemble ($\alpha$ = 0.5)| 44.30 |80.44|0.60|
> |Ensemble (non-constant $\alpha$)| 50.98|80.82|0.64|
> |TCP-SB w/o traj att ($\alpha$ = 0.3)|51.39|80.26|0.63|
> |TCP-SB w/o traj att ($\alpha$ = 0.5)|46.87|83.68|0.59|
>
> Yes, it does. As requested, we add new ablations for *direct ensemble* (2nd row in the table above), and *ensemble of multi-step control prediction and trajectory prediction w/o trajectory-guided attention* (last two rows in the table above). Our purpose is to provide a general and flexible fusion mechanism (L360). We **do not claim a constant $\alpha = 0.3$ and current turning-based criterion are optimal**. Sometimes $\alpha=0.5$ has similar performance to $\alpha=0.3$. It is reasonable since a large portion of steps belongs to non-turning situation, so the control branch is not utilized enough with a small $\alpha$. Thus, we design a sophisticated fusion rule to keep a non-constant $\alpha$ (3rd row in the table above). In this case, when *traj-specialized*, $\alpha$ is 0.5 for steer and throttle to utilize the control branch better, and the maximum brake value is taken. And $\alpha$ is set to 0.3 for all actions when *control-specialized*. We choose routes from different towns with the ones used for ablations from the CARLA leaderboard repo to determine this rule.

---

> > ### Comment · Reviewer_cv3D · 2022-08-03
> > **Response to rebuttal**
> >
> > I appreciate the additional ablations and clarifications provided by the authors.
> >
> > In A1, it is mentioned that the GRU in the trajectory branch can be replaced by MLPs to output all waypoints at once. This contradicts L182-183 which states that future waypoints are obtained in an auto-regressive fashion, inspired by [41]. Since an auto-regressive architecture is used, the waypoint output at each future timestep takes into account the prediction from the previous timestep. Can the authors provide some clarification about the GRU in the trajectory branch in Fig. 2? Is it similar to the waypoint prediction module of [41]? This would also help with a better understanding of `Traj-Only-multistep` and `MTL-2heads` architectures.
> >
> > The results in A2 and Tables 3, 4 indicate that the main performance gain comes from multi-step control prediction, shared backbone between trajectory & control branch, and ensemble. The gains from the situation-based fusion scheme ($\alpha=0.3$) and trajectory-guided attention seem marginal.

---

> > > ### Author Response · Authors · 2022-08-05
> > > **Author Response to Reviewer cv3D**
> > >
> > > Thanks for the follow-up discussion. We further address each concern to clarify technical details below.
> > >
> > > > In A1, it is mentioned that the GRU in the trajectory branch can be replaced by MLPs to output all waypoints at once. This contradicts L182-183 which states that future waypoints are obtained in an auto-regressive fashion, inspired by [41]. Since an auto-regressive architecture is used, the waypoint output at each future timestep takes into account the prediction from the previous timestep. Can the authors provide some clarification about the GRU in the trajectory branch in Fig. 2? Is it similar to the waypoint prediction module of [41]? This would also help with a better understanding of `Traj-Only-multistep` and `MTL-2heads` architectures.
> > >
> > > Yes, the GRU module to predict waypoints is similar to [41,10,27]. It is one implementation choice among many for the **policy head**. The policy head could be a GRU to auto-regressively obtain future waypoints one by one (L182-183), or MLPs to output all waypoints at once (A1). We choose GRU as the candidate,  as commonly does in [41,10,27].
> > >
> > > As we stated in A1, IMHO, it is not reasonable to directly add future feature loss to each step inside GRU since the output (four waypoints) is **only for the policy of current step**. The future feature is for future policy output. During rebuttal as requested, in order to add future feature loss to the trajectory branch, we change role of the original GRU to the temporal module, and use a MLP as the policy head.
> > >
> > > Specifically, the input to GRU is the concatenation of the current feature $\rm{j_t^{traj}=MLP(h_t^{traj})}$, a waypoint and the target point (c.f. original GRU input is a waypoint and the target point, as in [41,10,27]). This modification inherits from the temporal module design in the control branch (L209-211). In this case we treat a single waypoint as the policy output for one step, and we can add the future feature loss. The `Traj-Only-multistep` and `MTL-2heads` variants are designed based on this approach.
> > >
> > > > The results in A2 and Tables 3, 4 indicate that the main performance gain comes from multi-step control prediction, shared backbone between trajectory & control branch, and ensemble. The gains from the situation-based fusion scheme ($\alpha=0.3$) and trajectory-guided attention seem marginal.
> > >
> > >
> > > **Gain from situation-based fusion scheme.** As we stated in A2, **setting $\alpha$ to 0.3 $\neq$ our situation-based fusion scheme**. One can design a more tailored rule both for the choice of $\alpha$ and situaion criterion to achieve better performance (`Row2 vs Row3` in the Table of A2).
> > >
> > >
> > > **Gain from trajectory-guided attention.** Trajectory and control prediction are two closely related tasks which share common underlying representations while maintaining intrinsic differences. Thus it is reasonable to extract **shared features at an early stage** and have two interacted branches later (which is proposed in this work, `Row1` in the Table below). Separate backbones may lead to different feature representations and confuse the control branch when additional attention applies. IMHO, an interaction discarding the shared underlying representations (new experiment in rebuttal, `Row2` in the Table below) is inappropriate.
> > >
> > > ||attention gain(DS)|
> > > |:------------------------------------:|:------:|
> > > |shared backbone -> attention -> fusion (ablations in paper)|3.21 (Table3, R3 vs R4)|
> > > |fusion -> attention -> shared backbone (exp. in rebuttal)|1.07 (TCB-SB w/o att vs TCP-SB)|

---

> ### Author Response · Authors · 2022-08-02
> **Author response to Reviewer cv3D (2/2, Clarifications and Additional Suggestions)**
>
> **Q3: How does the multi-step control mitigate the IID assumption.**
>
> **A3:** As we discussed in **A1**, our multi-step control prediction aims to match the future latent features and action predictions with those from the expert. Our model has the ability to reason about what to-take action could match the future states with the expert, mitigating the IID assumption to some extent.
>
> **Q4: How does the trajectory branch help to incorporate static layout information (eg. curbs).**
>
> **A4:** The static layout information (lanes & curbs) needed for driving policy is tightly related to the location of the vehicle at a certain time step. As the trajectory branch predicts waypoints for each future step, it contains rich information about possible future ego locations. Therefore, the trajectory branch can help the control branch to focus on the important static information to better generate policy output at future steps.
>
> **Q5: Why does TCP has lower RC compared to LiDAR-based methods. Is it because TCP gets blocked more often, or taking the maximum of brake values leads to an overly conservative policy.**
>
>
> **A5:** In Table 2 in the Supplementary, TCP gets higher agent block penalty than [10,43]. As we discussed in Sec. 4.2, LiDAR-based methods have better object detection ability to avoid blocking. When the agent stops for a long time, they would move slowly if no obstacles detected ahead. An overly conservative policy is also possible in theory, but we do not observe such situation in our experiments.
>
>
> **Q6: Evaluation protocol on local validation is different than online leaderboard submission. The authors should include them in the paper. How is $\alpha$ selected and is there a separate validation protocol for tuning these values.**
>
> **A6:**  As we mention in L298, we use the same validation routes as LAV[10]. For the choice of $\alpha$, in order to achieve a better performance, we choose a non-constant $\alpha$ value over action types and situations for online leaderboard submission. We also discuss this in **A2**. This setting is chosen by analyzing the results on our validation routes. We will add these ellaborations accordingly.
>
> **Q7: If PID controllers could be tuned to better follow the trajectory.**
>
>
> **A7:** PID parameters and several thresholds are already carefully tuned in [15,41]. We follow the setting in [15]. In this rebuttal, we also tune the parameters of PID controllers in some validation routes provided from official CARLA leaderboard; chosen routes are from different towns compared with the ones for ablations. Specifically, we make two sets of PID parameters so that we can alter according to current situation (whether the vehicle is turning) similar to our fusion mechanism.
> Although parameters are tuned carefully and two sets of parameters are obtained, **the performance improvement is trivial (Driving Score from 28.29 to 30.63)**. This is because the environment for tuning is different from that for evaluation (e.g. road topology and curvature of turnings).
> This also **validates our motivation to avoid onerous parameter tuning** (which may still perform poorly in new environment).
>
> **Q8: Implementation details regarding the GRU modules, MLPs, and weights for loss terms.**
>
>
> **A8:** We have added details of the model structure, parameters of PID controllers, and loss weights in the revised Supplementary. Please check it in the **Implementation Details** section (Sec. B).
>
>
> **Q9: It would be helpful if the authors could provide comparison to FASNet[28]. This would greatly improve the paper.**
>
> **A9:** We do not exactly implement FASNet[28] due to limited time; however we address Reviewer's concerns below:
> 1. In the original FASNet, it has inferior performance compared with LBC[11]. But TCP surpasses LBC and its follow-up works including WoR[12] and LAV[10] by a large margin on the leaderboard.
> 2. Since FASNet is evaluated on `NoCrash` benchmark, we collect 6-hours data in Town01 under `NoCrash` training protocol (FASNet uses 100-hours), and train our Control-Only model and TCP model on it. We provide results on the most challenging setup of `NoCrash` benchmark: new town new weather with dense traffic in the table below. TCP is significantly better than FASNet.
>
> |NoCrash-New Town,New Weather,Dense|*FASNet[28]*|Control-Only|**TCP**|
> |:------------------------------------:|:------:|:------------:|:-------:|
> |Success Rate (%)|*32*|24|**60**|
>
> **Q10: Visualize GradCAM attention maps for multi-step control prediction branch without the trajectory-guided attention module.**
>
>
> **A10:** We have added visualization examples and discussions in the revised version of the Supplementary. Please check it in the **Experiments** section (Sec. C.3).
>
> **Relevent References**. We will add them in the revised manuscript.

---

> > ### Comment · Reviewer_cv3D · 2022-08-03
> > **Response to rebuttal**
> >
> > A3: The multi-step control prediction aims to match the state distribution between expert and policy but the training data would still be IID. I suggest that the argument about mitigating the IID assumption should be removed from Sec 3.2.2.
> >
> > A4: I agree that the future waypoints of the ego-vehicle are indicative of the presence of the road but their association with curbs and lanes is not clear. I suggest that this argument should be removed from L202-203.
> >
> > A9: It is hard to interpret the provided results since the CARLA versions are different (this is also the case for LBC comparison in the FASNet paper). FASNet is an important baseline because:
> > - it also predicts future actions for multiple timesteps (Sec 3.1)
> > - it also uses a recurrent architecture for future state prediction (Fig. 2)
> > - it also jointly learns waypoint and control prediction
> >
> > So, it is important to understand the differences in the capabilities of FASNet and TCP.

---

> > > ### Author Response · Authors · 2022-08-05
> > > **Author Response to Reviewer cv3D**
> > >
> > > Thanks for the follow-up discussion. We further address each concern to clarify technical details below.
> > >
> > > > The multi-step control prediction aims to match the state distribution between expert and policy but the training data would still be IID. I suggest that the argument about mitigating the IID assumption should be removed from Sec 3.2.2.
> > >
> > > We agree with Reviewer that each training sample is IID. We appreciate your suggestion and will revise accordingly.
> > >
> > > Please note that, since we predict multiple steps, and the temporal module involves previous action to predict the next one, corresponding expert data (GT) consists of continuous states and actions, which is not IID among the sequence. To this end, we wrote in the paper that it somehow mitigates the IID assumption.
> > >
> > > > I agree that the future waypoints of the ego-vehicle are indicative of the presence of the road but their association with curbs and lanes is not clear. I suggest that this argument should be removed from L202-203.
> > >
> > > Thanks for the suggestion. We agree that there are no quantitative associations, and we use "e.g., curbs and lanes" to give concrete examples about the intuition behind the guided attention module in L202-203. In fact, we could observe that curbs and lanes are salient parts in Fig. 4 to validate this qualitatively. We are pleased to take suggestions and revise accordingly.
> > >
> > > > It is hard to interpret the provided results since the CARLA versions are different (this is also the case for LBC comparison in the FASNet paper). So, it is important to understand the differences in the capabilities of FASNet and TCP.
> > >
> > > Agreed and thanks. Here we make some clarifications on differences/capabilities between FASNet and TCP. We will add these in the revised manuscript accordingly.
> > >
> > > - **Multi-step actions.** Previous predicted action has **no influence on the future states and control predictions** in FASNet. The future control action prediction of FASNet is mainly designed to take the weighted average (while we do not use the future prediction results during testing). The motivation and detailed approach are different from TCP.
> > > - **Recurrent architecture.** FASNet uses another video prediction PredNet [a] to predict/extrapolate future images. The recurrent architecture is inside PredNet, which **is pretrained without any interactions with the control prediction** (Sec. 4.1).
> > > - **Jointly learns waypoint and control.** FASNet predicts positions and headings of the vehicle **only as auxiliary tasks**, and these predictions **are utilized only at the training time** (Sec. 3.2). The trajectory branch and multi-step control branch in TCP have interactions in between, and the policy outputs from both branches are utilized and fused.
> > >
> > > [a] William Lotter et al. Deep Predictive Coding Networks for Video Prediction and Unsupervised Learning. ICLR, 2017.

---

### Official Review · Reviewer_6yad · 2022-07-11

**Rating:** 4
**Confidence:** 5
**Soundness:** 3 good
**Presentation:** 3 good
**Contribution:** 2 fair

**Summary:**

The paper studies how to combine image-to-trajectory and image-to-control approaches for autonomous driving in the Carla simulator. Both approaches have their advantages and disadvantages, by combining them the paper is able to achieve a new state of the art on the carla leaderboard only using a front facing camera. To fuse the temporal trajectories and static control predictions, the paper proposes to predict input trajectories which are feed through a temporal model similar to a world model that predicts future features and allows trajectory guided attention in the control branch.


**Questions:**

-Did you investigate DAgger to improve the driving of the proposed agent?
-Did you evaluate how well the temporal module can predict the future? For example by running the open loop action predictions in Carla or simulating a vehicle model?
-PID controllers are not well suited for lateral trajectory tracking since the are missing a "feedforward" component. However, there are several approaches which can deal with this, such as pure pursuit, curvature based feedforward, or as mentioned MPC. All of them are easy to implement, so I was wondering if using such a controller would change the finding, and result in a different conclusion (mainly trajectory predictions is all you need similar to some works done by Lyft/Woven Planet Level 5).

**Limitations:**

Both are discussed but only in the supplementary.

**Strengths And Weaknesses:**

Strengths: The paper is well written, clear to follow and achieves impressive results. I especially like the idea of the temporal module in the multi step control branch, which allows to match control inputs and input features.

Weaknesses: My main concern is that I am not sure if this paper fits NeurIPS? There are little technical contributions and interesting links such as the one between the temporal module and world models are not discussed. On the link to world models, the proposed training approach for the temporal module is actually know to not work well, since it is a purely deterministic model which is not able to capture the stochastic future. Potentially a hybrid approach such as RSSM could work better.

---

> ### Author Response · Authors · 2022-08-02
> **Author response to Reviewer 6yad (1/2)**
>
> Dear Reviewer 6yad,
>
> Thank you for your comments. We address your concerns on the weaknesses below.
>
> **Q1: Weakness: Interesting links between the temporal module and world models. The proposed training approach for the temporal module might not work well for world models. Potentially RSSM could work better.**
>
> **A1:** Thanks.
> The proposed temporal module shares some spirits with world models. World models [See references **a,b,c** below] formulate latent dynamic of the environment, which can then be used by **model-based RL methods.**
> In this work, we formulate the method in Imitation Learning (IL) domain; we do not aim to fully model the complex driving environment.
> The temporal module is designed to match future latent feature representations with the ones from the expert based on current feature and action.
> **Our policy head does not interact with the temporal module in the way where model-based RL interacts with world models.**
> Since the temporal module is differentiable, our model has the capability to reason about what current actions could match (i) future states and (ii) actions with the expert. To some extent, this mitigates IID assumption problem for the imitation learning task in an easier way.
>
> We might argue that our work is an IL-based method. Our temporal module spans a short horizon into the future (4 steps in our case) with direct supervision on future features. Different from world models, **a simple GRU module with deterministic states suffices to perform this task.**
>
>
> **Potential Hybrid Adaptation such as RSSM.** Agreed. In this rebuttal, we add discussions on RSSM in Dreamer [b], where the world model in [b] is built on a recurrent state space model [a].
> The world model in Dreamer servers as an environment to provide long-horizon trajectory, and it has to capture the stochastic nature of the world.
> It needs both stochastic and deterministic components. However, as world models are mainly utilized for reinforcement learning, how to devise and utilize a RSSM like world model for imitation learning for more complex environment (like autonomous driving) is an interesting topic to explore.
>
> We will add discussions about links between the temporal module and world models in the revised manuscript.
>
> ***If this paper fits NeurIPS***. Several pioneering works [d,e] (date back to the 1980's) to adopt neural network for end-to-end autonomous driving were published on NeurIPS. In recent years, there are also published work [f,g,h,i] in the NeurIPS main proceedings. IMHO, this work fits NeurIPS audience.
>
>
> > [a] Danijar Hafner et al. Learning latent dynamics for planning from pixels. ICML, 2019.
> >
> > [b] Danijar Hafner et al. Dream to control: Learning behaviors by latent imagination. arXiv, 2019.
> >
> > [c] David Ha et al. World models. NeurIPS, 2018.
> >
> > [d] Dean Pomerleau. Alvinn: An autonomous land vehicle in a neural network. NeurIPS, 1989.
> >
> > [e] Yann LeCun et al. Off-road obstacle avoidance through end-to-end learning. NeurIPS, 2007.
> >
> > [f] Yunpeng Pan et al. Learning Deep Neural Network Control Policies for Agile Off-Road Autonomous Driving. NeurIPS, 2017.
> >
> > [g] Andrew Spielberg et al. Learning-In-The-Loop Optimization: End-To-End Control And Co-Design of Soft Robots Through Learned Deep Latent Representations. NeurIPS, 2019.
> >
> > [h] Arthur Delarue et al. Reinforcement Learning with Combinatorial Actions: An Application to Vehicle Routing. NeurIPS, 2020.
> >
> > [i] David Acuna et al. Towards Optimal Strategies for Training Self-Driving Perception Models in Simulation. NeurIPS, 2021.

---

> ### Author Response · Authors · 2022-08-02
> **Author response to Reviewer 6yad (2/2)**
>
> **Q2: Incorporating DAgger to improve the driving of the proposed agent.**
>
> **A2:** Traditional DAgger needs policy output from the expert to be the same with that from the student model.
> In this work, the expert predicts the single step action at each step, while the student model requires trajectory and multi-step actions.
> These supervisions are not available since the expert does not interact with the environment when we adopt DAgger. Therefore, conventional DAgger can **not** be applied directly. In our preliminary experiments (not shown in the paper), we designed an approach to overcome the caveat in DAgger, by letting the expert take over the vehicle for certain steps to provide the required supervision.
> Results show that the refinement is feasible, but there are still many details to be determined or optimized, which is out of scope of this work. Thanks for the great advice.
>
>
> **Q3: How well the temporal module can predict the future. For example by running the open-loop action predictions in CARLA or simulating a vehicle model.**
>
> **A3:** As we do not reconstruct images from the temporal module, we provide the action and feature error for current and feature steps prediction on validation dataset below. From these new results, we conclude that TCP can generate satisfactory predictions with the proposed temporal module.
>
> ||Steer L1 Error|Throttle L1 Error|Brake L1 Error|Feature MSE|
> |:------------:|:--------------:|:-----------------:|:--------------:|:-----------------:|
> |Current Step|0.026|0.097|0.057|0.614|
> |Future Step1|0.029|0.139|0.113|0.633|
> |Future Step2|0.033|0.185|0.142|0.742|
> |Future Step3|0.035|0.209|0.151|0.914|
> |Future Step4|0.039|0.232|0.159|1.120|
>
>
> **Q4: Several approaches (pure pursuit/curvature-based feedforward/MPC) for lateral trajectory tracking; such a controller would change the finding and result in a different conclusion.**
>
> **A4:** Thanks for the comment. Note that PID parameters, the choice of the target angle for lateral control, and other hyperparameters are exquisitely tuned in [15,41] already. We follow the setting in [15].
> As requested, in rebuttal we provide experiments to use Pure Pursuit for lateral control.
> The experimental results are shown below.
>
> ||Driving Score|Route Completion|Infraction Penalty|
> |:--------------------------------------:|:-------------:|:----------------:|:------------------:|
> |Trajectory-Only (PID)|28.29|58.11|0.50|
> |Trajectory-Only (Pure Pursuit)|20.24|55.97|0.37|
> |Trajectory-Only (Situation-based PID)|30.63|66.13|0.53|
>
> > Experiment settings: We tune the parameters of PID controllers in the validation routes provided from official CARLA leaderboard;
> these routes are from different towns compared with the ones used for our ablation study.
> For `Trajectory-Only (Situation-based PID)`, we have two sets of PID parameters so that we can alter according to current situation
> (whether the vehicle is turning in this case) similar to the proposed fusion mechanism.
>
> One can observe that replacing `Trajectory-Only (PID)` with `Pure Pursuit` leads to **worse performance, from 28 degrading to 20.**
> Equipping with sophisticated controllers in the pipeline needs heavy engineering work, which is not the focus here.
> We would like to avoid the tuning process and keep the virtue of simplicity for end-to-end autonomous driving framework. "Trajectory planning is all you need similar to some work by Lyft" may be suitable in modular design where the planner has map information and accurate perception results.
>
>
> Nonetheless, we tune the parameters of PID controllers on other routes carefully and investigate if such an experiment would result in different conclusion. We can conclude that **the performance improvement is trivial** (`Situation-based PID` vs `PID`, 30 vs 28).
> This is because the environment for parameter tuning is different from that for evaluation (e.g. the road topology and the curvature of turnings).
> This new observation **validates the motivation to avoid onerous parameter tuning.** Besides, heuristic tuning may still perform poorly in the new scenarios.
>
>
> **Limitations.** We will shift the Limitation and Impact parts from Supplementary to Main paper in the revised manuscript.

---

### Official Review · Reviewer_X2A2 · 2022-07-11

**Rating:** 6
**Confidence:** 3
**Soundness:** 3 good
**Presentation:** 2 fair
**Contribution:** 3 good

**Summary:**

This paper presents a novel method for autonomous driving based on a single RGB camera on the vehicle. The method combines direct control commands from images into steering, and an indirect method with a policy that outputs local trajectories, which are then converted into steering commands with an analytical PID controller. The feature representation from the images is shared for both the trajectory and the direct control branches. The combination of commands is performed by a hard-coded heuristic method using a weighted mean based on two predefined situations: turning or not turning. The proposed approach provides good results on the CARLA benchmark, even compared to other methods that use more modalities or more RGB images as input.


**Questions:**

- How do you identify the type of driving mode to select a different fusing criterion?


**Limitations:**

- There is not much discussion about limitations, it would be good to have it. Also a clear evaluation of the error cases would be informative. What are the common failures of the solution?


**Strengths And Weaknesses:**

- Originality: the idea of combining a learned method for trajectory prediction and a learned method for motion control is novel in the domain of autonomous car driving. However, it has been explored for other applications in robotics, e.g., Pokle et al. “Deep Local Trajectory Replanning and Control for Robot Navigation”, 2019. I’d recommend extending the search to other application domains and include them in the related work. In any case, the paper brings novel ideas.
- Significance: the strength of this paper is in the evaluation. The method has been compared to other methods in a public benchmark, CARLA, and provided good results.
- Clarity: the paper is clearly written and it is easy to follow. There is some confusion with some terms, e.g., the final method is a combination of trajectory based control and direct end-to-end control, but the title seems to indicate something different.
- Quality: the paper has acceptable quality. The experiments are mainly the CARLA scores. I miss an important experiment: the fusing mechanism. The paper includes ablations using only trajectory-based or direct control, but not an evaluation of the importance of the hardcoded heuristic with the two types of driving cases. Even the information about how to classify in these two types is not clearly provided.

---

> ### Author Response · Authors · 2022-08-02
> **Author response to Reviewer X2A2**
>
> Dear Reviewer X2A2,
>
> Thank you for commenting on the strengths and contributions in the manuscript. We address Reviewer's questions each below.
>
>
> **Q1: Originality: In any case, the paper brings novel ideas; it's better to extend the search to other application domains and include them in the related work.**
>
> **A1:** Thank you for bringing up the paper, Pokle et al. in robotics; we will add in the Related Work section alongside with other potential applications.
>
>
> **Q2: Clarity: Confusion with the title. Final method is a combination but the title indicate otherwise.**
>
> **A2:** The title *Trajectory-guided Control Prediction* represents the way we fuse and obtain better feature representation from two branches (trajectory and control). The general idea is to introduce a novel end-to-end systematic philosophy for the ultimate task (controlling) in autonomous driving - whether it is a trajectory-guided control or/and end-to-end direct control.
> We are pleased to discuss with Reviewer and take suggestions about the title.
>
>
> **Q3: Quality and Questions: add ablation on the fusion mechanism -
> an evaluation to the hardcoded heuristic with two types of driving cases.
> How to identify the type of driving mode to select a different fusing criterion.**
>
>
> **A3:** Thanks. Though directly averaging two branches achieved good performance, we provide an example for more flexible and general scheme to fuse the two branches (L360). More tailored if-then schemes can be further designed as well. In our experiment, we choose **whether the vehicle is turning** as the criterion of the *situation* as we mention in L282-284. If the vehicle is turning (half of the steering actions within the last 1 second are larger than 0.1), the *situation* is *control specialized*, otherwise *trajectory specialized*. Under this criteria, we have conducted the experiments on different choices of the fusion weight $\alpha$ (aka heuristic parameter) in the **last part of Sec. 4.4** and presented in Fig. 5. The results on the importance of the fusion scheme is shown in the **last row in Table 3**.
>
>
>
> **Q4: Limitations. A clear evaluation of the error cases would be informative. What are the common failures.**
>
>
> **A4:** Thank you for the comments. As requested, we have provided additional examples and visualizations in the **Limitation** section of the Supplementary (due to space limit in the Main paper) in our revised version (Sec. D.1.1). In general, typical failure cases include:
>
> 1. Vehicles initially outside the ego agent's front view rush into the ego path with a high speed, causing a collision when emergent braking fails.
> 2. The ego agent fails to consider the possible trajectory of other vehicles, resulting in a blocking or collisions.
>
> **Analysis.** For the first case, it is because of the limited view of the monocular camera, hence a straightforward future direction is to add multi-view cameras or a LiDAR input to our agent. For the second case, the reason is that our model lacks the ability to reason about trajectories of other vehicles without an explicit prediction module, therefore another possible direction is to extend the multi-task learning framework with detection and motion prediction modules, and combine their results with our planned trajectory.

---

> > ### Comment · Reviewer_X2A2 · 2022-08-08
> > **Replies**
> >
> > Q2: The title is more of a personal thing but the message I get from your paper is that "Trajectory Guided Control Prediction" is one of the branches (the one with the waypoints), combined with pure reactive control, and that the combination is key. I think a title reflecting that would be more accurate, something like "Fusing Planning and Reactive Control for ...", but that is just my opinion, this is not a hard comment.
> >
> > Q3: Yes, I have seen the experiments on the fusion weight, but the unexplored part (and probably more critical, given the experiments with alpha values) is the "situation" detector. Seems like a very hard-coded rule in an otherwise learned approach
> >
> > Q4: thanks for including limitations

---

> > > ### Author Response · Authors · 2022-08-09
> > > **Author Response to Reviewer X2A2**
> > >
> > > Thanks for the follow-up discussion.
> > >
> > > > Q3: Yes, I have seen the experiments on the fusion weight, but the unexplored part (and probably more critical, given the experiments with alpha values) is the "situation" detector. Seems like a very hard-coded rule in an otherwise learned approach.
> > >
> > > Agreed. Developing a learning-based approach to involve the fusion part in the end-to-end training paradigm and replace the rule one is indeed an interesting and promising direction. A few methods could be explored towards this, e.g., (1) modeling these two branches with a probalistic uncertainty, (2) learning a discriminative model to score the degree of specialty of two branches respectively, (3) incorporating a gating network as the (Conditional) Mixture-of-Experts (MoE). Thank you for the suggestion and we will add this into the future work part.

---

### Author Response · Authors · 2022-08-02
**General Author Response for Rebuttal**

We thank Reviewers for helpful and detailed comments on our work.

TCP is a simple and yet effective vision-based solution for end-to-end autonomous driving framework. We get a unanimous agreement from all four Reviewers that "the idea is clear and presentation is easy to follow. The pipeline is novel." Most importantly, we achieve impressive result with a large improvement to the second best method on the public Carla benchmark leaderboard, with simple camera input alone.

We have added more ablative experiments in the rebuttal and clarify some technical details. Please see each response below. Thanks.

---

### Meta-Review · Area_Chair_EvuW · 2022-08-27

**Recommendation:** Accept
**Confidence:** Certain

**Metareview:**

The paper got split reviews: 1x reject, 1x borderline reject, 1x weak accept, 1x accept. All reviewers found the impressive performance on the challenging CARLA leaderboard to be a major strength of the paper.

Reviewer concerns stem from two factors: a) not enough technical contribution to warrant publication at NeurIPS (but results are still publication worthy at more domain-specific conferences eg ICRA, IROS), and b) bulk of the impressive performance (19 points) coming from the ensembling heuristic and only 6 points coming from proposed architectural modifications (shared backbone, multi-step control, temporal module and trajectory guided attention).

The meta-reviewer read through the paper, the reviews, the author response, and reviewer discussion. For the meta-reviewer, the impressiveness of the empirical results on a well-studied and important benchmark dominates the above reviewer concerns. As long as there is clear attribution and some understanding as to where this impressive performance improvement is coming from, the community will benefit from being aware of the results even though the proposed method may not be as technically deep as typical NeurIPS papers. The authors are encouraged to include the additional experiments conducted during the rebuttal phase into the final version of the paper, in particular the ones that help distill out the contribution of the different parts of the proposed system.

**Award:**

No

---

### Decision · Program_Chairs · 2022-09-14

Accept